# Potential of Biochar as a Peat Substitute in Growth Media for *Lavandula angustifolia*, *Salvia rosmarinus* and *Fragaria × ananassa*

**DOI:** 10.3390/plants12213689

**Published:** 2023-10-26

**Authors:** Giuseppina Iacomino, Alessia Cozzolino, Mohamed Idbella, Giandomenico Amoroso, Tomaso Bertoli, Giuliano Bonanomi, Riccardo Motti

**Affiliations:** 1Department of Agricultural Science, University of Naples Federico II, 80055 Naples, Italy; alessia.cozzolino2@unina.it (A.C.); mohamed.idbella@unina.it (M.I.); giandomenico.amoroso@unina.it (G.A.); giuliano.bonanomi@unina.it (G.B.); motti@unina.it (R.M.); 2Laboratory of Biosciences, Faculty of Sciences and Techniques, Hassan II University, Casablanca 28806, Morocco; 3BiokW, Piazza del Duomo 30, 38122 Trento, Italy; tomaso.bertoli@biokw.it; 4Task Force on Microbiome Studies, University of Naples Federico II, 80055 Portici, Italy

**Keywords:** climatic change, growing media, environmental sustainability, peat reduction, nursery production, wood vinegar

## Abstract

Peat has long been the primary substrate for the production of ornamental and horticultural plants in pots. Today, peat is no longer considered a renewable resource due to its very lengthy regeneration time. Biochar, a solid by-product of biomass pyrolysis, has been proposed as an agricultural soil amendment. We investigated the effects of two types of biochar, namely biochar from pruning wood waste and biochar activated with wood vinegar (“smoked biochar”), on two ornamental plants (*Lavandula angustifolia* and *Salvia rosmarinus*) and on strawberries (*Fragaria × ananassa*). For both types of biochar, we measured the following parameters: the pH, density, electrical conductivity, humidity, calcium carbonate, total carbon, nitrogen, potassium, calcium, magnesium, sodium, and water retention. For peat, we measured the following parameters: the pH, electrical conductivity, total carbon, and total nitrogen. Our results showed an overall increase in plant growth, particularly in *L. angustifolia* when using 10% and 50% biochar concentrations and a 10% concentration of biochar activated with wood vinegar. In *S. rosmarinus*, we observed a slight increase in the total plant weight with the application of 10% smoked biochar (biochar activated with wood vinegar). Finally, in *F. × ananassa*, we observed an increase in the plant weight and fruit production when 10% biochar was applied. On the other hand, when high concentrations of biochar (50% and 100%) and especially smoked biochar were applied, we observed a significant reduction in the growth of all plants. We concluded that biochar and biochar activated with wood vinegar showed remarkable biological activity with marked phytotoxicity at high concentrations. They promoted plant growth when applied diluted and their use as partial peat substitutes could help support more sustainable horticultural practices.

## 1. Introduction

Peat is partially decomposed organic material (mainly Sphagnum and bog plants) that has accumulated as a result of waterlogging, nutrient and oxygen deficiency, and high acidity [1]. Roughly twice as much carbon is stored in peatlands, which make up roughly 3% of the world’s land surface, as compared to global forest biomass [2]. Due to its advantageous physicochemical properties (high water holding and air capacity, absence of pathogens and weed seeds, low pH, bulk density, and nutrient content), peat is the most widely used substrate for growing media in the greenhouse and nursery industries [3]. However, due to rising peat prices and growing environmental concerns about the impact of peat extraction on peatland ecosystems, its use in the formulation of growing media has been limited, and environmentally friendly substrates have been sought [4]. Compost, derived from various agricultural by-products, such as sewage sludge and chicken and bovine manure, organic fibers and biochar have been tested as peat substitutes in the formulation of growing media for vegetables and ornamentals [5]. 

Biochar is a carbon-rich solid material formed through thermochemical transformation in the presence of limited available oxygen [6]. It is commonly used as a soil amendment in agroecosystems to enhance soil quality and increase productivity [7]. The primary advantages of using biochar as a soil amendment include the conservation of carbon reserves, improved water retention capacity, and the adsorption of phytotoxic organic molecules [8,9]. Recent research suggests that combining biochar with fresh organic material is a promising method for enhancing plant growth promotion effects [10]. This research interest stems from biochar’s potential for interacting with mineral nitrogen, leading to a retention that reduces volatilization and leaching and increases the nitrogen use efficiency [11]. This effect is attributed to the enhanced surface retention of ammonium on the biochar material [12]. For example, Shi et al. [13] observed a remarkable 14% increase in the overall fresh shoot weight and a substantial 25% increase in the fresh root weight when using a granular biochar–mineral urea composite compared to conventional urea fertilizer alone. However, most of the previous studies have focused on investigating the activation of biochar with nitrogen-rich organic amendments [14], undecomposed animal feces [15], or mineral fertilizer [16], with no studies examining the potential use of biologically derived products like wood vinegar. Wood vinegar, a nitrogen poor liquid material obtained by distilling the smoke produced during pyrolysis, has been shown to improve germination, growth, and the production of healthier plants [17]. Other studies have described the stimulatory effects of smoke on various wild plants and crops [18]. Moreover, recent studies have demonstrated the biological activity of wood vinegar against pests and pathogens [19] due to its low pH and high content of acetic acid and phenols [20]. 

In recent years, several studies have explored the potential of biochar for use in growing media for ornamentals and vegetables as an alternative to peat. To date, various studies have been conducted on the effects of biochar as a substitute for peat in containers for ornamental plants, yielding contrasting results [21,22,23,24]. Given the increasing need for sustainable and eco-friendly practices, finding alternatives to conventional substrates like peat is crucial. Peat extraction can have adverse effects on fragile ecosystems, such as wetlands and peatlands, resulting in habitat destruction and carbon dioxide emissions [25]. In this context, the primary objective of our study was to investigate two types of biochar: biochar produced from pruning wood waste (referred to as “biochar”) and the same biochar activated by the addition of wood vinegar (referred to as “smoked biochar”). We assessed the effects of biochar and smoked biochar as potential alternatives to peat and as substrates for the growth of *L. angustifolia*, *S. rosmarinus*, and *F. × ananassa* in terms of their shoot and root development. Rosemary, scientifically known as *S. rosmarinus*, is a popular herb that can be grown in pots or in the ground. Compost-based growing media are commonly used for cultivating herbs like rosemary, and the incorporation of biochar into these growing media can have several potential effects. *Lavandula* belongs to the Lamiaceae family, with *Lavandula angustifolia* Mill. and *L. dentata* L. being two of the most well-known and frequently cultivated species in the Mediterranean region. They have applications in the fragrance, cosmetic, and pharmaceutical industries. *F. × ananassa*, commonly referred to as strawberry, is a popular crop that thrives in a well-draining, nutrient-rich growing medium.

By investigating the potential use of biochar and smoked biochar as substitutes for peat, this study aims to contribute to the development of eco-friendly solutions for plant growth and cultivation. This, in turn, would aid in the preservation of fragile ecosystems, the reduction in carbon dioxide emissions, and the promotion of more sustainable agricultural practices. The specific objectives of our study included assessing the impact of different concentrations of biochar and smoked biochar mixed with peat on plant growth and establishing the correlations between the variations in the chemical composition of biochar and smoked biochar and their effects on plant growth. Such insights can prove valuable for understanding the mechanisms underlying the plant responses to these substrates. This knowledge can ultimately assist in optimizing the production and application of biochar, potentially leading to increased plant productivity and improved plant health. 

## 2. Results 

### 2.1. Biochars Chemistry 

The biochar was supplied by the company BiokW (Trento, Italy), which also provided us with all the chemical analyses relating to the biochar and the biochar activated with wood vinegar (i.e., smoked biochar). The density was slightly higher in the smoked biochar than in the biochar. The pH was slightly more acidic in the smoked biochar than in the biochar. The electrical conductivity and humidity were higher in the smoked biochar. The total limestone, however, was higher in the biochar. Carbon and the total nitrogen were present in higher quantities in the smoked biochar. Potassium, calcium, magnesium, and sodium were more present in the biochar than in the smoked biochar. The biochar also had a higher total ash concentration and a higher water retention than the smoked biochar. The carbon to hydrogen ratio was the same for both types of biochar. The calculated particle size fraction was slightly higher in the biochar. The peat was acidic, and the electrical conductivity (EC) showed higher values than the biochar, but close to those of the smoked biochar. Finally, the total carbon was lower than the two biochars, while the total nitrogen was slightly higher than the biochar and the smoked biochar (Table 1).

### 2.2. Salvia rosmarinus

The Department of Agricultural Sciences’ botanical garden was used to establish the rooted cuttings of *S. rosmarinus* spenn. and *L. angustifolia*, while Salvi Vivai S.S. of Ferrara, Italy provided the strawberry plants (*F. × ananassa* (duchesne ex Weston) Duchesne ex Rozier) for the plant testing. A total of 105 1.5 L pots were filled with three different biochar application rates—10%, 50%, and 100%—by mixing the biochar with peat as the substrate. When necessary, distilled water was added to each pot and the pots were then placed in the greenhouse from March to June 2022. In order to calculate the plant biometrics, the dry weight and the relative distribution of dry biomass were measured at the end of the four-month production cycle for each of the three species. 

In the pots containing the 10% smoked substrate biochar, the *S. rosmarinus* plants exhibited a significant weight increase as compared to the plants cultivated in the pots with 100% peat (control). However, no significant difference was observed in the pots with 10% biochar when compared to the control. Conversely, a notable decrease in the plant weight was observed in all other tested concentrations when compared to the control (Figure 1A). With respect to the biomass of the roots and leaves, no significant difference was observed in the pots treated with various concentrations of biochar and smoked biochar, except for a marginal increase in the root biomass allocation in the pots with pure biochar and smoked biochar (Figure 1B). Figure 1C shows the *S. rosmarinus* plants at the beginning of the experiment and after four months of growth in the presence of biochar and smoked biochar at different dosages.

### 2.3. Lavandula angustifolia 

The *L. angustifolia* plants recorded a statistically significant growth increase in the pots containing 10% and 50% biochar and 10% smoked biochar when compared to the pots with pure peat (control). Conversely, the other concentrations exhibited comparable outcomes, except for 100% smoked biochar, which recorded a much lower growth than the control (Figure 2A). Furthermore, a notable increase in the shoot biomass allocation was observed, particularly in the pots with 10%, 50%, and 100% biochar and 10% smoked biochar (Figure 2B). Figure 2C shows the *Lavandula angustifolia* plants at the beginning of the experiment and after four months of growth in the presence of biochar and smoked biochar at different dosages.

### 2.4. Fragaria × ananassa 

For the *F. × ananassa* plants, a significant difference in the weight was observed only for the pots with 10% biochar compared to the pots with pure peat (Figure 3A). It was noteworthy that both types of biochar, applied at 50% and 100%, caused a significant reduction in growth compared to the control. Regarding the biomass distribution, no significant differences were observed compared to the control, except for smoked biochar, which was applied at 50% and 100% and caused a higher root allocation (Figure 3B). Finally, a significant increase in the fruit yield was reported in the pots treated with 10% biochar compared to pots treated with 100% peat, and in all the pots treated with biochar and smoked biochar, except for pots treated with 100% smoked biochar (Figure 3C). Figure 3D shows the *F. × ananassa* plants at the beginning of the experiment and after four months of growth in the presence of biochar and smoked biochar at different dosages.

## 3. Discussion

In the current study, we explored the potential of biochar as a substitute for peat in compost-based growing media for two ornamental plants (*L. angustifolia* and *S. rosmarinus*) and strawberries (*F. × ananassa*) cultivated in pots within a greenhouse setting. Our findings revealed varying effects that were contingent upon the plant species and the concentrations tested. In the case of *S. rosmarinus*, the application of 10% smoked biochar displayed a subtle stimulating effect when compared to the plants grown in substrates composed solely of 100% peat. Conversely, in *L. angustifolia*, both biochar and 10% smoked biochar exhibited a significant stimulating impact. Lastly, in the context of *F. × ananassa*, the utilization of 10% biochar led to a stimulation in both the total plant weight and fruit weight compared to the substrates consisting entirely of 100% peat. 

In relation to *S. rosmarinus*, our results indicated that high concentrations of biochar activated with wood vinegar (50% and 100%) had evident phytotoxic effects, whereas stimulatory or neutral effects were observed at lower concentrations (10%). In this context, Bonanomi et al. [26] previously reported that smoke waters prepared from various raw materials exhibited remarkable biological activities and potential applications for promoting plant growth. These substances tend to have phytotoxic effects at high concentrations but display neutral or stimulatory effects at lower concentrations. Our findings aligned with this pattern, showing a similar response of plant species to wood vinegar-activated biochar, namely, reduced growth at higher tested concentrations and increased growth at lower concentrations. One potential mechanism underlying the concentration-dependent plant responses was the presence of chemical inhibitors in the smoke water that promoted root growth at lower concentrations, a phenomenon known as hormesis [27]. The presence of karrikins, a chemically defined family of plant growth regulators first isolated from burned plant tissue smoke, has been linked to the stimulatory effects of smoke products (e.g., smoke, aerosol smoke, and smoke water) on seed germination and root and shoot growth [28]. Karrikins have triggered seed germination in numerous plant species and enhanced seedling growth. Additionally, our chemical analyses revealed a higher electrical conductivity in the biochar activated with wood vinegar. This could suggest the presence of potentially phytotoxic soluble substances, including acetic acids and phenols, which can elevate the electrical conductivity in smoked biochar. 

Utilizing biochar as a substitute for peat in compost-based growing media for *S. rosmarinus* holds the potential for enhancing plant growth and promoting sustainability. However, it is crucial to carefully consider the specific type and application rate of biochar to achieve optimal results. It is important to note that the effects of biochar on *S. rosmarinus* and other plants may vary depending on the factors, such as the type and quantity of biochar used, as well as the specific growing conditions. Before making general recommendations, conducting field trials and collecting data on the performance of biochar-based growing media in various contexts is essential.

Our findings regarding *L. angustifolia* aligned with those of Fascella et al. [29], who conducted a study on the differences in growth, ornamental quality, and nutrient distribution in two lavender species (*L. angustifolia* and *L. dentata*). They were cultivated in a peat-based growing medium with varying volumes of wood biochar (0%, 25%, 50%, or 75%). In their study, as in ours, a higher percentage of biochar in the substrate had a detrimental impact on the growth of potted plants. However, lavender grown with 25% biochar exhibited comparable growth and qualitative responses concerning the parameters, such as the number of leaves, root length, biomass production, and biomass water use efficiency, when compared to the plants cultivated with 0% biochar. Furthermore, in both studies, the stimulating effect of biochar activated with wood vinegar was consistently observed when diluted to 10%, but it displayed significant phytotoxicity at higher concentrations. Validating our results on containerized lavenders helped elucidate the relationship between the concentration of biochar in the growing medium, its fundamental characteristics, and the plant’s response, encompassing morphological, physiological, and biochemical aspects. This encourages the utilization of biochar for sustainable nursery production of potted ornamentals, offering favorable outcomes from both economic and environmental perspectives, including the recycling of agricultural by-products and the reduction in peat usage.

Numerous studies have explored the effects of biochar on the growth of *F. × ananassa*, but this study represents the first attempt to assess biochar’s potential as a replacement for peat-based substrates. It is noteworthy that, in this study as well, the most favorable performance was consistently observed at an application rate of 10%. A study conducted by De Tender [30] examined the impact of biochar as an additive on the soil and substrate physicochemical properties, plant growth, disease susceptibility, and rhizosphere microbiology in strawberries cultivated in white peat. The addition of 3% biochar to the peat resulted in an increase in the fresh and dry plant weight. Biochar-induced alterations in soil microbiology have been widely observed, but the exact mechanisms behind these microbiological changes remain elusive [31]. Several hypotheses have been put forth. Biochar might offer an additional habitat for bacteria and fungi [32], provide shelter for fungivores [33], disrupt intercellular communication among microbes [34], and influence the composition of the microbiota. Organic compounds from biochar have been shown to inhibit certain members of the microbiome while promoting others [35]. Moreover, biochar can modify the physicochemical properties, such as the pH and electrical conductivity (EC), which in turn can affect the microbiome [36]. 

In our study, the incorporation of 10% biochar resulted in an increased total plant weight and fruit weight, supporting the notion that biochar can function as a biostimulant [31] and can serve as an excellent substitute for peat at low concentrations. Additionally, biochar may exert its influence on plant growth by altering the rhizosphere microbiome, potentially increasing the availability of plant nutrients like nitrogen, thus promoting plant growth [37]. Ultimately, the chemical analyzes that were carried out highlighted some relevant factors of biochar to support its role as an ideal plant growth substrate. Indeed, the application of biochar can influence various soil properties, including the pH, bulk density, cation exchange capacity, water retention, and biological activity. Biochar reduces leaching and nutrient loss through volatilization by altering the soil pH and improving the ion exchange capacity [38]. Biochars generally have a pH range between 6.52 and 12.64; in our case the value was 7.9 for biochar and 7.5 for smoked biochar. The application of alkaline biochar tends to increase the pH of acidic and neutral soils [39] and this could further confirm the ability of biochar to neutralize acidic substrates, such as peat in our case.

It is evident that multiple factors are at play in the biochar’s impact on the plant–soil system. The physicochemical composition of the soil or substrate, plant development, disease resistance, and microbial populations in the plant–soil/substrate system can all be influenced by the presence of biochar. We recommend that future research endeavors concentrate on comprehensively examining how biochar affects the soil, plants, and the microbial flora within the system simultaneously. In light of these complexities, biochar holds promise in agriculture, but is contingent upon the specific medium or culture substrate system employed.

## 4. Materials and Methods

### 4.1. Study Site 

The experiment was carried out at the Department of Agricultural Science research station located in the Royal Park of Portici (40°48′40.3″ N, 14°20′33.8″ E; 75 m a.s.l.), at the foot of the south-western slopes of Vesuvius Mountain in the province of Naples (Southern Italy). The study site was an area dedicated to the cultivation of vegetables in greenhouses and in open fields. With an annual mean temperature of 16.2 °C and monthly mean temperatures ranging from 25.9 °C in August to 9.1 °C in January, the study area has a typical Mediterranean climate. The annual rainfall is 929 mm, with most of the rainfall occurring during the winter (290 mm), spring (200 mm), and autumn (348 mm) and with a significant dry season in the summer (89 mm). 

### 4.2. Experimental Design

Two types of biochar were used: biochar produced from pruning wood waste (“biochar”), and activated biochar that was obtained by soaking 1 kg of dry biochar in 1000 mL of wood vinegar for 48 h (“smoked biochar”). The wood vinegar was produced through the pyrolysis process, carried out in a plant in Ferrara (North Italy), which uses woody pruning waste from pears and apple orchards. The woody pruning wastes, after being freed from the soil particles and crushed into pieces less than 10 cm long, are subjected to slow pyrolysis until they reach 800 °C. During pyrolysis, the gases are fed into a steel tube where they condense, and the pyrolytic liquid, here called wood vinegar, is collected in a plastic container at the bottom. After decanting, the liquid has a brown-orange color (Figure 4). The wood vinegar was acidic (pH 3.2) and had an electrical conductivity of 1389.10 μS/cm. The phenol concentration was 54.00 mg/mL, and the acetic acid was 27840.16 mg/mL, corresponding to 2.78%. 

The plant tests were performed with rooted cuttings of *S. rosmarinus* Spenn. and *L. angustifolia* grown in the botanical garden of the Department of Agricultural Sciences and with strawberry plants (*F. × ananassa* (Duchesne ex Weston) Duchesne ex Rozier) supplied by SALVI VIVAI S.S. (Ferrara, Italy). Three biochar application rates, i.e., 10%, 50%, and 100% (calculated in v/v proportion), were prepared by mixing biochar with peat in volume to fill a total of 105 pots at 1.5 L. The biochar supplied by the company was air dried at room temperature before use in the pots experiment. The peat and the two biochars types had very low moisture contents (<5%) at the timing of the pot preparation. The pots were then placed in the greenhouse from March to June 2022 and irrigated to field capacity as needed by adding distilled water to each container. At the end of the four-month production cycle for all three species, the plant biometrics were quantified by measuring the dry weight and the relative distribution of the dry biomass. The measurements were carried out with the aid of a precision balance (OHAUS Precision Standard–Model TS400D). The experimental design resulted in a total of 35 pots for each species (three biochar application rates × two biochar types × five replicates + five control replicates with peat only).

### 4.3. Biochar and Peat Chemical Analyses

The biochar was supplied by the company BiokW (Trento, Italy). The chemical analyzes were carried out for the following parameters: pH [40], density [41], electric conductivity [42], humidity [41], total limestone (method: Dlgs 7276 del 31/05/16 Suppl. 13 n. 2), total carbon (Dlgs 7276 del 31/05/16 Suppl. 13 n. 2), ashes 550 °C [43], molar ratio H:C (method: Dlgs 7276 del 31/05/16 Suppl. 13 n. 2), total nitrogen [44], total potassium, total calcium, total magnesium, total sodium [45], water retention (method: DM 1/08/97 SO n. 173 GU 204 2/09/1997 Met.4), and particle size fraction <5.00 mm, <2.00 mm, and <0.50 mm [46]. In regard to the peat, we evaluated the pH, electrical conductivity, total carbon, and total nitrogen. The total carbon was determined according to the Italian directive (G.U. n.21 del 26/01/2001), while the total nitrogen was determined according to the European directive (Regalement CE n.2003 of 13/10/2003). For the determination of the EC and pH, the official procedures of the Italian National Society of Soil Science were used [47].

### 4.4. Statistical Analysis

The significance of variation in the plant biomass and fruit production between treatments was evaluated using the ANOVA test, and the means were separated pairwise using the Tukey post hoc test to provide further details on the level of significance between the treatments. The level of significant differences was assessed using *p* < 0.05. All the analyses were performed using the STATISTICA 13.3 software.

## 5. Conclusions

Peat is now considered a non-renewable resource due to its slow regeneration. Our study showed that biochar can serve as a partial peat substitute for ornamental as well for some horticulture species. In detail, biochar and smoked biochar can promote plant growth, especially in *L. angustifolia* and *F. × ananassa* when applied at lower concentrations (10% and 50%). However, caution should be exercised with high concentrations of biochar (50% and 100%) and biochar activated with wood vinegar, as they resulted in significant reductions in plant growth in all the studied species. Therefore, it is critical to identify the proper biochar application rate that should be mixed with peat or other organic substrates. Considering these results, we suggest that biochar, particularly when applied at a relatively low application rate (10%), holds great potential as a partial replacement for peat in horticulture and floriculture productions.

## Figures and Tables

**Figure 1 plants-12-03689-f001:**
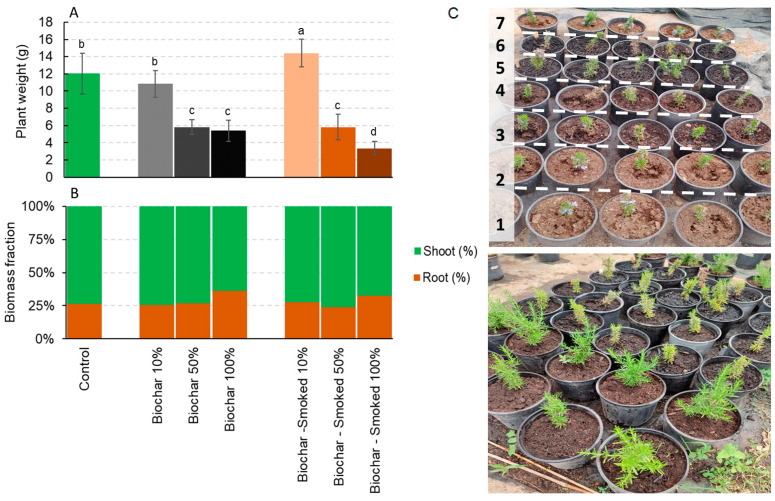
The total weight (**A**) and biomass allocation (**B**) of *Salvia rosmarinus* after four months of growth in the control and in the presence of biochar and smoked biochar at different dosages. (**C**) *Salvia rosmarinus* at the beginning of the experiment and after four months of growth in the presence of biochar and smoked biochar. The different numbers indicate, respectively, (1) smoked biochar 10%, (2) biochar 10%, (3) smoked biochar 50%, (4) biochar 50%, (5) smoked biochar 100%, (6) biochar 100%, and (7) the control. The values are the average of eight replicates ± the standard deviation. The different letters indicate statistically significant differences (*p* < 0.05, Duncan test). The grey color and brown color scale refer to the application rate of biochar and smoked biochar, respectively.

**Figure 2 plants-12-03689-f002:**
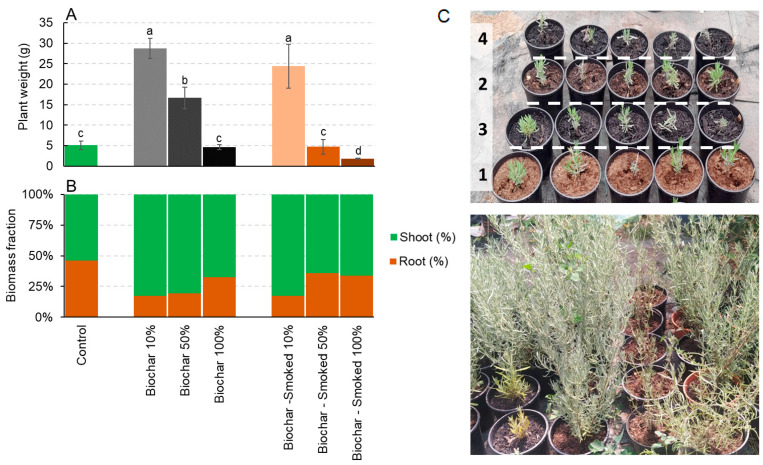
The total weight (**A**) and biomass allocation (**B**) of *Lavandula angustifolia* after four months of growth in the control and in the presence of biochar and smoked biochar at different dosages. (**C**) *Lavandula angustifolia* at the beginning of the experiment and after four months of growth in the presence of biochar and smoked biochar. The different numbers indicate, respectively, (1) the control, (2) biochar 10%, (3) biochar 50%, and (4) biochar 100%. The values are the average of eight replicates ± the standard deviation. The different letters indicate statistically significant differences (*p* < 0.05, Duncan test). The grey color and brown color scale refer to the application rate of biochar and smoked biochar, respectively.

**Figure 3 plants-12-03689-f003:**
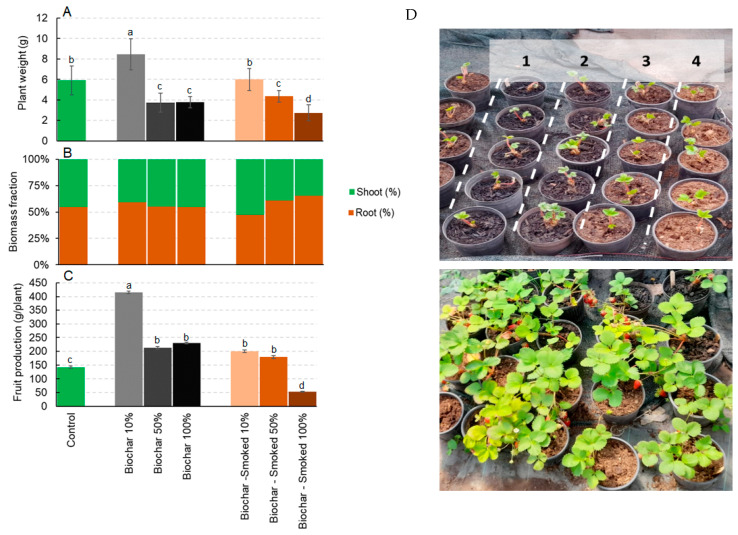
The total weight (**A**), biomass allocation (**B**) and fruit production (**C**) of *Fragaria × ananassa* after four months of growth in the control and in the presence of biochar and smoked biochar at different dosages. (**D**) *Fragaria × ananassa* at the beginning of the experiment and after four months of growth in the presence of biochar and smoked biochar. The different numbers indicate, respectively, (1) smoked biochar 100%, (2) biochar 100%, (3) smoked biochar 50%, and (4) the control. The values are the average of eight replicates ± the standard deviation. The different letters indicate statistically significant differences (*p* < 0.05, Duncan test). The grey color and brown color scale refer to the application rate of biochar and smoked biochar, respectively.

**Figure 4 plants-12-03689-f004:**
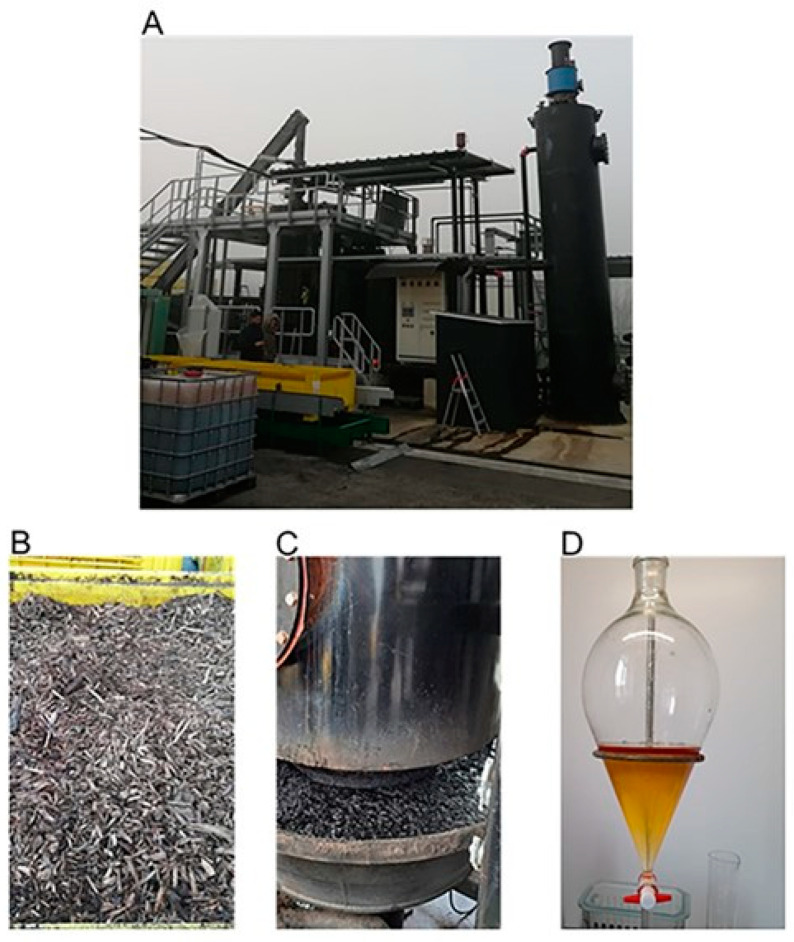
Pyrogasifier (**A**), the organic feedstock consisting in wood debris from orchards (**B**), the obtained biochar (**C**), and the wood vinegar used to drench the biochar (**D**).

**Table 1 plants-12-03689-t001:** List of the main chemical characteristics of biochar, biochar—smoked and peat.

Parameters	Unit	Biochar	Smoked Biochar	Peat
Density	g/L	438	457	
pH	pH unit	7.9	7.5	5.70
Electric conductibility	mS/m	29	49	42.38
Humidity	% m/m	59.2	60.3	
Total limestone (CaCO_3_)	% s.s.	7.1	5.2	
Total carbon	% s.s.	55.7	70.7	44.50
Ashes 550 °C		26.51	19.04	
Molar ratio H:C		0.29	0.29	
Total nitrogen	% s.s.	0.49	0.64	1.09
Total potassium	% s.s.	0.92	0.76	
Total calcium	% s.s.	2.95	2.38	
Total magnesium	% s.s.	0.43	0.31	
Total sodium	mg/kg s.s.	1258.00	884.24	
Water retention	% m/m	74.85	73.53	
Particle size fraction < 5.00 mm	% m/m s.s	87	77	
Particle size fraction < 2.00 mm	% m/m s.s	47	44	
Particle size fraction < 0.50 mm	% m/m s.s	25	24	

## Data Availability

The authors confirm that the data supporting the findings of this study are available within the article.

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
