# Peer review of "Potential of Biochar as a Peat Substitute in Growth Media for Lavandula angustifolia, Salvia rosmarinus and Fragaria × ananassa"

_plants, 2023, doi:10.3390/plants12213689_

Round 1
Reviewer 1 Report
In this manuscript (plants-2616430) entitled "Potential of biochar as peat substitute in growth media for Lavandula angustifolia, Salvia rosmarinus and Fragaria × ananassa" submitted to Plants, Giuseppina Iacomino and colleagues investigated the effect of two types of biochar, namely biochar from pruning wood waste and biochar activated with wood vinegar ("biochar-smoked") on two ornamental plants (Lavandula angustifolia and Salvia rosmarinus) and on strawberries (Fragaria x ananassa). Through careful analyses, authors conclude that biochar and biochar activated with wood vinegar showed remarkable biological activity with marked phytotoxicity at high concentrations, while it promoted plant growth when applied diluted and its use as a partial peat substitute could help support more sustainable horticultural practices. The data are convincing and the writing is clear and straightforward. However, some issues need to be addressed for improving the quality of this manuscript.
1, For Figure 1, pictures of Salvia rosmarinus after four months of growth in the control and in the presence of biochar and smoked-biochar at different dosages should be included in the revised Figure 1. In addition, the shoot height and root length of Salvia rosmarinus should be analyzed in the revision.
2, For Figure 2, pictures of Lavandula angustifolia after four months of growth in the control and in the presence of biochar and smoked - biochar at different dosages should be included in the revised Figure 2. In addition, the shoot height and root length of Lavandula angustifolia should be analyzed in the revision.
3, For Figure 3, pictures of Fragaria × ananassa after four months of growth in the control and in the presence of biochar and smoked - biochar at different dosages should be included in the revised Figure 3. In addition, the shoot height and root length of Fragaria × ananassa should be analyzed in the revision.
4, For Figure 4B and 4C, scale bars should be included in the revised pictures.
Author Response
In this manuscript (plants-2616430) entitled "Potential of biochar as peat substitute in growth media for Lavandula angustifolia, Salvia rosmarinus and Fragaria × ananassa" submitted to Plants, Giuseppina Iacomino and colleagues investigated the effect of two types of biochar, namely biochar from pruning wood waste and biochar activated with wood vinegar ("biochar-smoked") on two ornamental plants (Lavandula angustifolia and Salvia rosmarinus) and on strawberries (Fragaria x ananassa). Through careful analyses, authors conclude that biochar and biochar activated with wood vinegar showed remarkable biological activity with marked phytotoxicity at high concentrations, while it promoted plant growth when applied diluted and its use as a partial peat substitute could help support more sustainable horticultural practices. The data are convincing and the writing is clear and straightforward. However, some issues need to be addressed for improving the quality of this manuscript.
ANSWER: We thank the reviewer for his valuable comments and try to satisfy his requests. Please, check the specific responses for details.
1, For Figure 1, pictures of Salvia rosmarinus after four months of growth in the control and in the presence of biochar and smoked-biochar at different dosages should be included in the revised Figure 1. In addition, the shoot height and root length of Salvia rosmarinus should be analyzed in the revision.
ANSWER: Images of Salvia rosmarinus after four months of growth in control and in the presence of biochar and smoked biochar at different dosages have been included in the revised Figure 1, as requested by the reviewer. In addition, we added the following sentence to line 130: “Figure 1C shows Salvia rosmarinus plants at the beginning of the experiment and after four months of growth in the presence of biochar and biochar-smoked at different dosages”.
Data on shoot height and root length of Salvia rosmarinus, Lavandula angustifolia and Fragaria were not collected, since we wanted to focus our attention on the influence of biochar and biochar-smoke on plant weight and total root biomass and shoot. In fact, plant height for such species is nor very relevant and auto-correlated with plant biomass.
2, For Figure 2, pictures of Lavandula angustifolia after four months of growth in the control and in the presence of biochar and smoked - biochar at different dosages should be included in the revised Figure 2. In addition, the shoot height and root length of Lavandula angustifolia should be analyzed in the revision.
ANSWER: Images of Lavandula angustifolia after four months of growth in control and in the presence of biochar and smoked biochar at different dosages have been included in the revised Figure 2, as requested by the reviewer. In addition, we added the following sentence to line 152: “Figure 2C shows Lavandula angustifolia plants at the beginning of the experiment and after four months of growth in the presence of biochar and biochar-smoked at different dosages”.
3, For Figure 3, pictures of Fragaria × ananassa after four months of growth in the control and in the presence of biochar and smoked - biochar at different dosages should be included in the revised Figure 3. In addition, the shoot height and root length of Fragaria × ananassa should be analyzed in the revision.
ANSWER: Images of Fragaria × ananassa after four months of growth in control and in the presence of biochar and smoked biochar at different dosages have been included in the revised Figure 3, as requested by the reviewer. In addition, we added the following sentence to line 178: “Figure 3D shows Fragaria × ananassa plants at the beginning of the experiment and after four months of growth in the presence of biochar and biochar-smoked at different dosages”.
4, For Figure 4B and 4C, scale bars should be included in the revised pictures.
ANSWER: Figure 4 is intended to be an example and representative of the way in which wood vinegar is produced. The woody pruning waste shown in figure 4B are examples and can have variable dimensions, while figure 4B shows examples of biochar fragments less than 10 cm long, produced following the pyrolysis process.
Please see the attachment.

Reviewer 2 Report
In the manuscript “Potential of biochar as peat substitute in growth media for Lavandula angustifolia, Salvia rosmarinus and Fragaria × ananassa” by Lacomino et al., author have investigated the effect of two types of biochar on two ornamental plants (Lavandula angustifolia and Salvia rosmarinus) and on strawberries (Fragaria x ananassa), biochars chemistry, plant growth performance of the three plants and fruit production in strawberry were analyzed, the research results have certain significance for the cultivation practice of horticultural plants. However, there are still some issues that need to be explained reasonably, as follows:
1. In the INTRODUNTION, author mentioned that “In recent years, several studies have explored the potential of biochar for use in growing media for ornamentals and vegetables as an alternative to peat. ”, Lavandula angustifolia and Salvia Rosmarinus are the ornamental plants, but strawberry belong to perennial herbaceous fruit trees. Why did the author choose strawberries and two ornamental plants to conduct research together? What is the purpose of choosing Strawberry?
2. In the Materials and Methods, the biochar chemical analyses, only just briefly mentioned what method to use, is it entirely based on experimental guidance? Didn't you make any modifications in the determine process? It is recommended to briefly describe the steps and operations of the measurement.
3. How plant growth indicators are measured needs to be described in the material method.
4. In the RESULT, the data on plant height has an error bar, but there is no error bar for biomass and strawberry yield. Since conducting analysis of variance, all data should be analyzed, not just a portion. Even in biomass data, there is no indication of significance, so it is recommended to add it.
5. Figure 3, fruit production, what does it mean, the number of fruits, the total yield, or other indicators? The author needs to explain clearly.
6. The conclusion should summarize the research results of this article in the most concise sentences. Some background and theoretical statements should not appear in the conclusion. Please simplify the handle and rewrite this section.
7. Please check all references and follow the journal's requirements for layout.
Fluent English writing and good language proficiency.
Author Response
In the manuscript “Potential of biochar as peat substitute in growth media for Lavandula angustifolia, Salvia rosmarinus and Fragaria × ananassa” by Lacomino et al., author have investigated the effect of two types of biochar on two ornamental plants (Lavandula angustifolia and Salvia rosmarinus) and on strawberries (Fragaria x ananassa), biochars chemistry, plant growth performance of the three plants and fruit production in strawberry were analyzed, the research results have certain significance for the cultivation practice of horticultural plants. However, there are still some issues that need to be explained reasonably, as follows:
1.In the INTRODUNTION, author mentioned that “In recent years, several studies have explored the potential of biochar for use in growing media for ornamentals and vegetables as an alternative to peat”, Lavandula angustifolia and Salvia Rosmarinus are the ornamental plants, but strawberry belong to perennial herbaceous fruit trees. Why did the author choose strawberries and two ornamental plants to conduct research together? What is the purpose of choosing Strawberry?
ANSWER: We would like to thank the reviewer for his valuable comments and try to answer his concerns. The main aim of our research was to look for a valid alternative to the use of peat, a substrate widely used in greenhouses and nurseries, which unfortunately is causing considerable concern due to its environmental impact. The choice to also test strawberry plants, in addition to ornamental ones, was made to understand whether the use of biochar and smoked biochar as an alternative substrate was also useful for this type which production is moving from soil to soilless substrates in many areas. In this regards, identify proper substrate for soilless cultivation is important for Fragaria cultivation. The results obtained relating to strawberries brought added value to the results already obtained with ornamental as well for some horticulture plants, confirming the hypothesis that biochar and smoked biochar can represent valid alternatives to the use of peat, paving the way for expanding the research to other plants.
- In the Materials and Methods, the biochar chemical analyses, only just briefly mentioned what method to use, is it entirely based on experimental guidance? Didn't you make any modifications in the determine process? It is recommended to briefly describe the steps and operations of the measurement.
ANSWER: Chemical analyses were performed by performing the original methods without modification.
3.How plant growth indicators are measured needs to be described in the material method.
ANSWER: We have added the following clarifying sentence in the materials and methods: “The measurements were carried out with the aid of a precision balance (OHAUS precision Standard - Model TS400D)”, line 307.
- In the RESULT, the data on plant weight has an error bar, but there is no error bar for biomass and strawberry yield. Since conducting analysis of variance, all data should be analyzed, not just a portion. Even in biomass data, there is no indication of significance, so it is recommended to add it.
ANSWER: Ok, in the new version we add error bar for plant weight and production (yield). We don’t add error bars only in the plot reporting biomass fraction because the graph does not allot it. The same apply to figures 1 and 2.
- Figure 3, fruit production, what does it mean, the number of fruits, the total yield, or other indicators? The author needs to explain clearly.
ANSWER: We meant the total yield of fruit, added to line 175: “Finally, a significant increase in fruit yield was reported in pots treated with 10% biochar compared to pots treated with 100% peat, and in all pots treated with biochar and biochar-smoked, except for pots treated with 100% biochar-smoked (Figure 3C)”.
- The conclusion should summarize the research results of this article in the most concise sentences. Some background and theoretical statements should not appear in the conclusion. Please simplify the handle and rewrite this section.
ANSWER: We have rewritten the conclusions trying to summarize and simplify the results as suggested by the reviewer, as follows: “Peat is now considered a non-renewable resource due to its slow regeneration. Our study showed that biochar can serve as partial peat substitute for ornamental as well for some horticulture species. In detail, biochar and biochar-smoked can promote plant growth, especially in L. angustifolia and F. x ananassa when applied at lower concentrations (10% and 50%). However, caution should be exercised with high concentrations of biochar (50% and 100%) and both biochar and biochar activated with wood vinegar, as they resulted in significant reductions in plant growth in all species studied. Therefore, it is critical to identify the proper biochar application rate to be mixed with peat or other organic substrates. Considering these results, we suggest that biochar, particularly when applied at relatively low application rate (10%), holds great potential as a partial replacement for peat in horticulture and floriculture productions.”
- Please check all references and follow the journal's requirements for layout.
Done.
Please see the attachment.

Reviewer 3 Report
The study discusses an important theme of substituting non-renewable resources in agriculture with more accessible ones. The experiment design is clear and simple, the manuscript is written very accurately. But some questions should be addressed before publishing to make the text even more clear. One of the most prominent ones are (1) why 10, 50, 100 % of biochar were chosen, (2) why didn’t you measure chemical characteristics of peat and compare it with biochar? Below are more specific questions:
1. Abstract
Some of the specific terms are introduced in the abstract and are not explained properly for the reader, unfamiliar with this theme:
Line 26 - It is not very clear, what do 10% and 50% mean in this context.
Line 27 – What does “biochar-smoked” stand for?
2. Introduction is nice, but some things require clarification:
Line 41 – The sentence “Peatlands represent about 3% of the world's land area but store about twice as much carbon as the world's forest biomass” almost completely repeats the first sentence in the abstract of the cited paper. Is it appropriate?
Line 65 – the term “activation of biochar” is introduced too briefly without emphasis.
Line 67 – Do you imply that “animal feces” are not “biologically derived products”?
Line 68 – Does wood vinegar also act as nitrogen-rich organic amendment?
All in all, I had to make additional research myself in order to understand that wood vinegar is a liquid smoke and what are its contents.
3. Results
Line 109 – Since methods section is at the end of the manuscript, the beginning of the results paragraph should explain a little, what is going on. Did you perform replicate measurements for biochar chemical characteristics? Do any results have any statistically significant difference?
Line 122 – no mentioning of the experiment set up.
Figure 1, 2, 3 – No explanation of brown and grey colours.
4. Discussion
The discussion lacks interpretation of obtained results through the lens of chemical characteristics of the used substrates.
5. Materials and methods
Is it common to grow these plants in pure peat? You mention that «Peat was once the primary component in ornamental plant substrates». So, shouldn’t usage of pure peat affect plant growth in some way too? What were the peat chemical characteristics? I think it is essential to implement them along with the characteristics of biochar in order to understand their effect on plant growth.
Line 270, Experiment design. Did peat, biochar and smoked biochar all have the same moisture content? How did you maintain the application rates of these substrates?
Line 287 – How did you chose 10% as the most appropriate rate for biochar usage? Why not 5, 15, 20%?
Line 289 – what was the field capacity for this experiment?
As for a non-native englisch speaker, the manuscript is writtent quite good. May be some articles are missing.
Author Response
The study discusses an important theme of substituting non-renewable resources in agriculture with more accessible ones. The experiment design is clear and simple, the manuscript is written very accurately. But some questions should be addressed before publishing to make the text even more clear. One of the most prominent ones are (1) why 10, 50, 100 % of biochar were chosen, (2) why didn’t you measure chemical characteristics of peat and compare it with biochar? Below are more specific questions:
ANSWER: We thank the reviewer for his comments and try to answer his concerns. As regards the first question, we decided to test biochar at 10, 50, 100%, to evaluate the impact of different concentrations of biochar mixed with peat on plant growth, choosing a wide range of concentrations and trying to understand which of it might be the best choice to use. Our results highlighted how low concentrations of biochar (10%) can promote plant growth and therefore be used as a partial substitute for peat, paving the way for the development of further research focused on the use of low concentrations of biochar. Regarding the second question, it was not our main objective to compare the chemical characteristics of peat with those of biochar. Peat is a scarce and largely non-renewable natural resource; therefore, peat substitutes are needed. The chemical-physical characteristics of biochar are well known, as are many of its benefits once inserted into the soil. For this reason, we tried to focus on which concentrations of biochar could bring benefits to the plant compared to the control with peat alone, to find a valid alternative to the use of peat.
- Abstract
Some of the specific terms are introduced in the abstract and are not explained properly for the reader, unfamiliar with this theme:
Line 26 - It is not very clear, what do 10% and 50% mean in this context.
ANSWER: We specified on line 26 that 10% and 50% biochar refer to the tested concentration: “Our results show an overall increase in plant growth, particularly in L. angustifolia when using 10% and 50% biochar concentration and 10% concentration of biochar activated with wood vinegar”.
Line 27 – What does “biochar-smoked” stand for?
ANSWER: Biochar-smoked stands for biochar activated with wood vinegar, definition added to line 28, “In S. rosmarinus, we observed a slight increase in total plant weight with the application of 10% biochar-smoked (biochar activated with wood vinegar)”.
- Introduction is nice, but some things require clarification:
Line 41 – The sentence “Peatlands represent about 3% of the world's land area but store about twice as much carbon as the world's forest biomass” almost completely repeats the first sentence in the abstract of the cited paper. Is it appropriate?
ANSWER: We changed the sentence as follows: "Roughly twice as much carbon is stored in peatlands, which make up roughly 3% of the world's land surface, as compared to global forest biomass.", added to line 41.
Line 67 – Do you imply that “animal feces” are not “biologically derived products”?
Answer: No, we mean in this specific study cases that “animal feces” was used. In the revised version we add the term “undecomposed” before animal feces.
Line 68 – Does wood vinegar also act as nitrogen-rich organic amendment?
Answer: No, Wood vinegar is an aqueous liquid generally comprised of acetic acid, methanol, furfural, phenol, acetaldehyde, allyl alcohol, furan, formic, propionic and butyric acid, and other volatile organics. Due to the vinegars' high levels of acetic acid concentration, which have been utilized successfully without causing any discernible environmental consequences, wood vinegar can be used as an organic pesticide, herbicide, and fungicide. To clarify this point, we add the terms “nitrogen poor” when describing wood vinegar.
- Results
Line 109 – Since methods section is at the end of the manuscript, the beginning of the results paragraph should explain a little, what is going on. Did you perform replicate measurements for biochar chemical characteristics? Do any results have any statistically significant difference?
Answer: We clarified by adding the following sentence to line 109: “The biochar was supplied by the company BiokW (Trento, Italy), which also provided us with all the chemical analyzes relating to the biochar and the biochar activated with wood vinegar (i.e., Biochar Smoked)."
Line 122 – no mentioning of the experiment set up.
Answer: We have added the following information to the line 125: “The Department of Agricultural Sciences' botanical garden was used to establish root-ed cuttings of S. rosmarinus Spenn. and L. angustifolia, while SALVI VIVAI s.s. of Ferrara, Italy provided strawberry plants (F. x ananassa (Duchesne ex Weston) Duchesne ex Rozier) for the plant testing. A total of 105 1.5 L pots were filled with three different biochar application rates—10%, 50%, and 100%—by mixing the biochar with peat as the substrate. When necessary, distilled water was added to each pot as the pots were then placed in the greenhouse from March to June 2022. In order to calculate plant biometrics, dry weight and the relative distribution of dry biomass were measured at the end of the four-month production cycle for each of the three species”.
Figure 1, 2, 3 – No explanation of brown and grey colours.
Answer: Ok, we integrated the captions as follows: “Grey color and brown color scale refer to the application rate of biochar and biochar-smoked, respectively”.
- Discussion
The discussion lacks interpretation of obtained results through the lens of chemical characteristics of the used substrates.
Answer: We have added additional information regarding chemical characteristics, as requested by the reviewer, to line 277: “Ultimately, the chemical analyzes carried out highlighted some relevant factors of biochar to support its role as an ideal plant growth substrate. Indeed, the application of biochar influences various soil properties including pH, bulk density, cation exchange capacity, water retention and biological activity. Biochar reduces leaching and nutrient loss through volatilization by altering soil pH and im-proving ion exchange capacity [38]. Biochars generally have a pH range between 6.52 and 12.64, in our case the value was 7.9 for biochar and 7.5 for smoked biochar. The application of alkaline biochar tends to increase the pH of acidic and neutral soils [39], this could further confirm the ability of biochar to promote plant growth, neutralizing highly acidic substrates, such as peat.”
- Materials and methods
Is it common to grow these plants in pure peat? You mention that «Peat was once the primary component in ornamental plant substrates». So, shouldn’t usage of pure peat affect plant growth in some way too? What were the peat chemical characteristics? I think it is essential to implement them along with the characteristics of biochar in order to understand their effect on plant growth.
Answer: The objective of our work was to evaluate the effect of biochar as a more sustainable alternative to the use of peat in terms of physical plant support. In this regards, basic characteristic of peat in term of nutrient availability is not very relevant because peat is very nutrient poor and is used as physical support in pot cultivation. In this type of cultivation, the plant nutrient requirement is satisfied by application of proper amount of mineral fertilizer. Based on such considerations, in this work we have not evaluated the chemical characteristics of peat, but we consider the referee suggestion for future works.
Line 270, Experiment design. Did peat, biochar and smoked biochar all have the same moisture content? How did you maintain the application rates of these substrates?
Answer: The moisture of the peat and biochar was very low (below 5%). The three biochar application rates, i.e., 10%, 50%, and 100%, were prepared by mixing with peat in volume to fill a total of 105 pots of 1.5 L, as indicated at line 304. Concerning moisture, we add this sentence: “Peat and the two biochar types have very low moisture content (<5%) at the timing of pot preparation”.
Line 287 – How did you chose 10% as the most appropriate rate for biochar usage? Why not 5, 15, 20%?
Answer: We chose the 10% concentration with the aim of evaluating the entire application range of biochar, starting from a low concentration up to the maximum usable concentration. Since the results clearly suggest that 10% biochar concentration was the most optimal, it will also be possible in the future to evaluate with more detail lower application rates i.e. 1%, 3%, 5%, 10%.
Line 289 – what was the field capacity for this experiment?
Answer: The field capacity for this experiment predicted a total of 105 pots of 1.5 L volume.

Round 2
Reviewer 2 Report
Dear editor, I have carefully reviewed this article and the author has made a lot of modifications according to the review comments. I think it can be published.Author Response
Please see the attachment.

Reviewer 3 Report
The authors addressed the comments of all three reviewers, but I'm sorry to admit that I don't think that this experiment is suitable for publishing in the Plants Journal.
1. The authors highlight that their idea was to find a substitution for peat and used this as an excuse to omit the evaluation of its chemical characteristics. However, in the discussion they also added a point that biochar can neutralise acidic substrates, such as peat. They could easily confirm this statement by their research, but alas, they “tried to focus on which concentrations of biochar could bring benefits to the plant compared to the control with peat alone, to find a valid alternative to the use of peat”. They didn’t add the discussion about actual obtained chemical characteristics, their differences between biochar types. I don’t see what this data adds to the research in this current form.
2. Authors declared that “Peat and the two biochar types have very low moisture content (<5%) at the timing of pot preparation”. But table 1 clearly states that the humidity of both biochars was around 60%. How is it possible? Another serious flaw is that since authors showed that biochar was advantageous for plant growth only at 10%, I think that even the 5% moisture content brings a serious alteration to the concentration of applied biochar. Did you measure w/w, v/w or v/v concentration?
3. Pictures of plants are nice, but it is not clear which experiment variants are shown.
In the end, authors declare that biochar is suitable for the substitution of peat, but it isn’t. They still have 90% (or we actually don’t know how much) of peat in the mixture. May be all the good effect of biochar comes from the pH alteration, which we actually don’t know, since the authors didn’t measure it. It seems that instead of biochar they could have used dolomite flour and be good with it.
Ok
Round 3
Reviewer 3 Report
The authors have addressed all the questions about the manuscript.
It is necessary for the manuscript to mention that % values were calculated in v/v proportion, not w/w. Because it can seriously alter proportion of the compounds with different dencity.
No comment